# Tracr: Compiled Transformers as a Laboratory for Interpretability

**David Lindner**[†]
Google DeepMind

**János Kramár**
Google DeepMind

**Sebastian Farquhar**
Google DeepMind

**Matthew Rahtz**
Google DeepMind

**Thomas McGrath**
Google DeepMind

**Vladimir Mikulik**[†]
Google DeepMind

## Abstract

We show how to "compile" human-readable programs into standard decoder-only transformer models. Our compiler, `Tracr`, generates models with known structure. This structure can be used to design experiments. For example, we use it to study "superposition" in transformers that execute multi-step algorithms. Additionally, the known structure of `Tracr`-compiled models can serve as *ground-truth* for evaluating interpretability methods. Commonly, because the "programs" learned by transformers are unknown it is unclear whether an interpretation succeeded. We demonstrate our approach by implementing and examining programs including computing token frequencies, sorting, and parenthesis checking. We provide an open-source implementation of `Tracr` at https://github.com/google-deepmind/tracr.

## 1 Introduction

Large language models (LLMs) are powerful but their inner workings are poorly understood (Danilevsky et al., 2020). The development of techniques for interpreting them is held back by a lack of *ground-truth* explanations (Yang et al., 2019). Our "compiler", `Tracr`, converts human-readable programs written in RASP, a domain-specific language for transformer computations (Weiss et al., 2021), into standard decoder-only transformers.

`Tracr` constructs models with known computational structure, which makes it easier to conduct interpretability experiments. As an example, we study neural networks' ability to compress a large number of sparse features into fewer dimensions using superposition (Elhage et al., 2022a). Compressing `Tracr` models using gradient descent allows us to study superposition in transformers implementing multi-step algorithms.

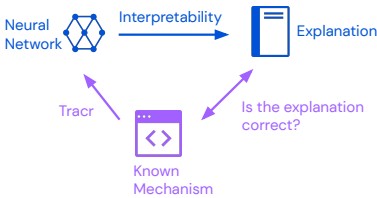

Figure 1: Interpretability tools produce explanations for a given neural network. `Tracr creates models` that implement a known mechanism. We can then compare this mechanism to explanations an interpretability tool produces.

A second use of transformers that implement known computations is evaluating interpretability methods aiming to reveal facts about a model's computation. `Tracr` could allow future work to directly test methods including, for example, classifier probes (Belinkov, 2022), gradient-based attribution (Nielsen et al., 2022), and causal tracing (Meng et al., 2022).

---

[†]Correspondence to dlindner@google.com, vmikulik@google.com.

37th Conference on Neural Information Processing Systems (NeurIPS 2023).

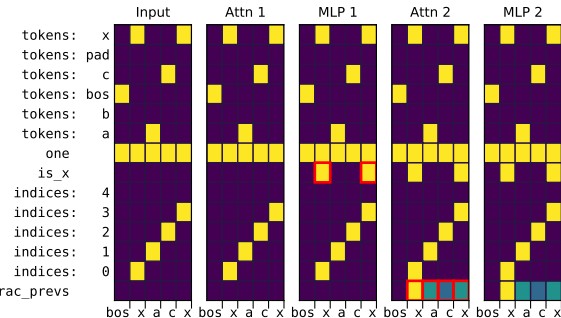

```
is_x = (tokens == "x")
prevs = select(indices, indices, <=)
frac_prevs = aggregate(prevs, is_x)
```

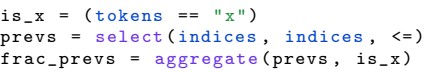

Figure 2: An example RASP program (left) that computes the fraction of previous "x" tokens at each position of the input. `Tracr` compiles this program to a transformer model. (right) A visualisation of a forward pass through the compiled model, showing the contents the residual stream, one panel per layer. The y-axis shows the residual stream dimensions, while the x-axis of each panel corresponds to the input sequence, "<bos>xacx" (x-axis). Changes to the residual are marked in red. Attn 1 is a no-op, MLP 1 computes the indicator variable `is_x`, Attn 2 implements the select-aggregate operation to compute `frac_prevs`, and MLP 2 is a no-op again. Section 4 discusses this and other examples in more detail. A detailed, step-by-step interpretation of the figure is provided in Appendix C.

Our main contributions are to: (1) Introduce `Tracr`, which "compiles" RASP programs into transformer models (Section 3); (2) Showcase models produced by `Tracr` (Section 4); (3) Provide a case-study where we examine superposition `Tracr` models compressed using gradient descent (Section 5). We confirm key observations by Elhage et al. (2022b) in a new setting: compressed models drop unnecessary features, and represent less important features in superposition.

In addition to aiding interpretability research, we think compiled models are a powerful didactic tool for developing a more concrete imagination for transformer mechanisms.

We discuss the applications and limitations of `Tracr` in Section 7 and Appendix A, and we provide an open-source implementation of the compiler at `https://github.com/google-deepmind/tracr`.

## 2 Background

This section provides an overview of key concepts our work builds on. In particular, we review the transformer model architecture and the RASP programming language.

### 2.1 Transformer Architecture

A transformer model consists of alternating *multi-headed attention* (MHA) and *multi-layer perceptron* (MLP) layers with residual connections. Multi-headed attention (Vaswani et al., 2017) computes attention maps on sequences of length $N$. A single attention head $i$ first computes an attention pattern

$$A^i = \text{softmax}\left((xW_Q^i)(xW_K^i)^T / \sqrt{d_k}\right) \in \mathbb{R}^{N \times N}$$

for some input $x \in \mathbb{R}^{N \times d}$, where $W_Q^i, W_K^i \in \mathbb{R}^{d \times d_k}$ are learnable parameters. Usually, we call the entries of $(xW_K^i)$ *keys*, and the entries of $(xW_Q^i)$ *queries*. *Multi-headed* attention combines $H$ attention heads heads by computing

$$\text{MHA}(x) = \text{Concat}\left[A^1(xW_V^1), \ldots, A^H(xW_V^H)\right] W_O$$

where $W_V^i \in \mathbb{R}^{d \times d_v}$ and $W_O \in \mathbb{R}^{Hd_v \times d}$ are another set of learnable parameters. We commonly call the entries of $(xW_V^i)$ *values*.

The MLP layers in transformer models compute $\text{MLP}(x) = \sigma(xW_1)W_2$ where $W_1 \in \mathbb{R}^{d \times h}$, $W_2 \in \mathbb{R}^{h \times d}$ are learnable weights, and $\sigma$ is a non-linear function; for simplicity, we use the Rectified Linear Unit (ReLU). In this paper, we focus on decoder-only transformers, which consists of alternating blocks of MHA and MLP.

When designing `Tracr`, we adopt the *transformer circuits* perspective, introduced by Elhage et al. (2021). This view (1) focuses on the transformer being a residual stream architecture and (2)

introduces an alternative parameterisation for attention operations. Taking this viewpoint, simplifies reasoning about the transformer architecture and will help us when assembling transformers manually.

**The residual stream view.** Transformers have residual connections at each attention and MLP layer. Elhage et al. (2021) consider the residual connections a core feature of the architecture and describe the model in terms of a *residual stream* that each layer reads from and writes to in sequence. The residual stream acts as a type of memory that earlier layers can use to pass information to later layers.

**Parameterising attention as** $W_{QK}$ **and** $W_{OV}$**.** Following Elhage et al. (2021), we parameterise an attention head by two (low-rank) matrices $W_{QK}{}^i = W_Q^i (W_K^i)^T / \sqrt{d_k} \in \mathbb{R}^{d \times d}$ and $W_{OV}{}^i = W_V^i W_O^i \in \mathbb{R}^{d \times d}$ where we split $W_O$ into different heads, such that $W_O = [W_O^1, \dots W_O^H]$, where each $W_O^i \in \mathbb{R}^{d_v \times d}$. We can then write MHA as

$$A^i = \text{softmax}\left(x W_{QK}{}^i x^T\right) \qquad \text{MHA}(x) = \sum_{i=1}^{H} A^i x W_{OV}{}^i$$

Importantly, we can think of MHA as summing over the outputs of $H$ independent attention heads, each parameterised by low-rank matrices $W_{QK}$ and $W_{OV}$. $W_{QK}$ acts as a bilinear operator reading from the residual stream, and $W_{OV}$ is a linear operator both reading from and writing to the residual stream. The softmax is the only nonlinearity in an attention head.

## 2.2 RASP

The "Restricted Access Sequence Processing Language" (RASP) is a human-readable computational model for transformer models introduced by Weiss et al. (2021). RASP is a sequence processing language with two types of variables, *sequence operations* (s-ops) and *selectors*, and two types of instructions, *elementwise* and *select-aggregate* transformations.

**Sequence operations.** A sequence operation (s-op) represents sequences of values during evaluation. `tokens` and `indices` are built-in primitive s-ops that return a sequence of input tokens or their indices, respectively. For example: `tokens("hello")` $= [\text{h}, \text{e}, \text{l}, \text{l}, \text{o}]$, and `indices("hello")` $= [0, 1, 2, 3, 4]$. S-ops roughly correspond to the state of the residual stream in transformers.

**Elementwise operations.** RASP allows arbitrary elementwise operations on s-ops. For example, we can compute `(3*indices)("hello")` $= [0, 3, 6, 9, 12]$. Elementwise operations roughly correspond to MLP layers in transformers.

**Select-aggregate operations.** To move information between token positions, RASP provides *select-aggregate* operations which roughly correspond to attention in transformers. A *selector* has a graph dependency on two s-ops and evaluates on inputs of length $N$ to a binary matrix of size $N \times N$. To create a selector, the `select` operation takes two s-ops and a boolean predicate $p(x, y)$. For example:

$$\texttt{select(indices, [1, 0, 2], <)("abc")} = \begin{bmatrix} 1 & 0 & 0 \\ 0 & 0 & 0 \\ 1 & 1 & 0 \end{bmatrix}.$$

Here, $p(x, y) = x < y$, where $x$ comes from `indices`, and $y$ comes from the constant s-op $[1, 0, 2]$.

The `aggregate` operation takes as input a selector and an s-op, and produces an s-op that averages the value of the s-op weighted by the selection matrix. For example:

$$\texttt{aggregate}\left(\begin{bmatrix} 1 & 0 & 0 \\ 0 & 0 & 0 \\ 1 & 1 & 0 \end{bmatrix}, \ [10, 20, 30]\right) = [10, 0, 15].$$

A selector roughly corresponds to an attention pattern in a transformer. Together a select-aggregate operation roughly corresponds to an attention head in transformers.

### 2.3 Mechanistic Interpretability and Superposition

Mechanistic interpretability (Cammarata et al., 2020; Olah, 2022) aims to produce mechanistic explanations of the inner workings of ML programs. This includes attempts to reverse engineer how neural networks implement specific behaviours (Cammarata et al., 2020).[3]

In Section 5, we study *superposition*: the ability of a neural network to approximately represent many more features than the number of dimensions of the embedding space (Elhage et al., 2022a). Despite preliminary evidence that superposition occurs in neural networks, it remains poorly understood, in part because it has only been studied in small (2-layers or less) networks that implement very simple algorithms (Elhage et al., 2022b; Scherlis et al., 2022). Understanding superposition in larger models could represent a major step forward for mechanistic interpretability (Olah, 2022).

## 3 `Tracr`: A Transformer Compiler for RASP

In this section, we provide an overview of how `Tracr` translate RASP programs to transformer weights. For more details on the implementation, we refer to Appendix D and our open-source implementation at `https://github.com/google-deepmind/tracr` including the accompanying documentation.

`Tracr` comes with an implementation of RASP embedded in Python. A RASP program is an expression graph which is incrementally constructed from atomic RASP operations. We make a few technical modifications to allow translating RASP to model weights: we disallow boolean combinations of selectors, enforce annotated categorical or numerical embeddings for the residual stream, and enforce the use of a beginning-of-sequence token. We discuss the motivations for each of these changes in Appendix B, where we also explain how any RASP program can be refactored to be compatible with these restrictions. In practice, we can implement programs to solve all tasks described by Weiss et al. (2021).

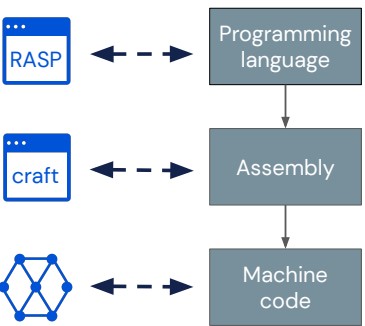

Figure 3: `Tracr` translates RASP to `Craft` and then to model weights, analogous to how programming languages are first translated to assembly then to machine code.

If RASP is the high-level language we compile, `Craft` is our "assembly language", offering slightly more abstraction than pure weight matrices (cf. Figure 3). `Craft` provides a transformer implementation using vector spaces with labelled basis dimensions and operations on them. This lets us define projections or other linear operations in terms of basis direction labels, which simplifies constructing model components that act on different vector spaces. As a bonus, models represented in `Craft` are independent of specific transformer implementations. Models compiled by `Tracr` can be translated into weights of any standard decoder-only transformer model (without layer norm).

`Tracr` translates RASP programs to transformer weights in six steps:

1. Construct a computational graph (Figure 4(a)).
2. Infer s-op input and output values (Figure 4(a)).
3. Independently translate s-ops into model blocks (Figure 4(b)).
4. Assign components to layers (Figure 4(c)).
5. Construct the model (Figure 4(c)).
6. Assemble weight matrices.

Let us go through these step by step. Figure 4 gives a schematic overview using an example program.

**1. Construct a computational graph (Figure 4(a)).** First, we trace the whole program to create a directed graph representing the computation. The graph has source nodes representing `tokens` and `indices` and a sink node for the output s-op. Each operation in the RASP program becomes a node in the computational graph.

---

[3]Our approach is complementary: we *construct* neural networks to implement specific behaviours.

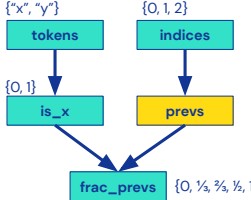 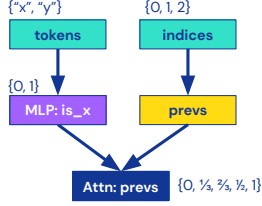 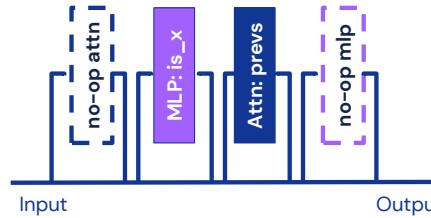

(a) Steps 1 & 2: Graph with inferred s-op value sets.

(b) Step 3: Nodes translated to MLPs and attention heads.

(c) Steps 4 & 5: Nodes allocated to locations in a model.

Figure 4: Schematic overview of how `Tracr` compiles the `frac_prevs` program from Figure 2 with a input vocabulary {"x", "y"} and context size 3. (a) shows the computational graph with value annotations after step 2 of the compilation. (b) shows how `is_x` and `frac_prevs` are translated to model components independently in step 3. (c) shows the assembled model which has two no-op components because models blocks always need to have one attention and one MLP layer.

**2. Infer s-op values (Figure 4(a)).** For each s-op, we need to decide how to embed it in the residual stream. To use categorical encodings, we need to know which values an s-op can take. All nodes have a finite set of output values because computations are deterministic, and we have a finite input vocabulary and context size. Therefore, in the second step, we traverse the graph and annotate each node with its possible outputs. This annotation uses simple heuristics that ensure we find a superset of the values an s-op will take, though, sometimes, an output set can contain values that the s-op never takes in practice.

**3. Independently translate s-ops (Figure 4(b)).** Next, we consider each node in the computational graph independently and translate it into a model block. Elementwise operations become MLP blocks, and select-aggregate operations become attention blocks. We use a library of manually engineered MLP and attention blocks to approximate arbitrary functions for numerical and categorical inputs and outputs. MLPs with categorical inputs and outputs function as lookup tables. MLPs with numerical inputs and outputs use piecewise linear approximations. For attention layers, we translate a selector into the $W_{QK}$ operator and the corresponding aggregate operation into the $W_{OV}$ operator. We only support attention with categorical inputs. We also do a few basic simplifications of RASP programs at this stage. For example, we combine consecutive elementwise operations into a single s-op. For more details on the MLP and attention blocks, see Appendix D.

**4. Assign components to layers (Figure 4(c)).** To construct a transformer model, we need to allocate all model blocks in the computational graph to layers. Ideally, we want to find the smallest model to perform the desired computation. We can generally formulate this as a combinatorial optimization problem with several constraints: the transformer architecture has alternating attention and MLP layers, and all computations that depend on each other need to be in the correct order. For scope reasons, we solve this problem heuristically. First, we compute the longest path from the input to a given node. This path length is an upper bound for the layer number to which we can allocate the node. Then we apply additional heuristics to combine layers with blocks that we can compute in parallel. This approach returns a correct but sometimes suboptimal layer allocation.

**5. Construct the model (Figure 4(c)).** We construct the residual stream space as the direct sum of all model components' input and output spaces. In other words, we embed each s-op in its own orthogonal subspace, which is reserved for its sole use throughout the entire network. Now, we can traverse the computational graph in the order determined by the layer allocation and stack the components to obtain a full transformer represented in `Craft`.

**6. Assemble weight matrices.** Finally, we translate the `Craft` representation of the model into concrete model weights. First, we combine parallel MLP layers into a single layer and parallel attention heads into a single layer. In attention layers, we then factor the $W_{QK}$ and $W_{OV}$ matrices into separate $W_q, W_k, W_o, W_v$ weight matrices. Finally, we adjust the shapes of all weights and connect them to our transformer architecture. We can then infer the model configuration (depth, layer width, residual stream size, etc.) to fit the elements we have created.

We use a standard decoder-only transformer implementation in Haiku (Hennigan et al., 2020), notably removing layer norms. Extending `Tracr` to support any other transformer implementation is straightforward by reimplementing only step 6.

```
smaller = select(tokens, tokens, <=)
target_pos = selector_width(smaller)
sel_sort = select(target_pos, indices, ==)
sort = aggregate(sel_sort, tokens)
```

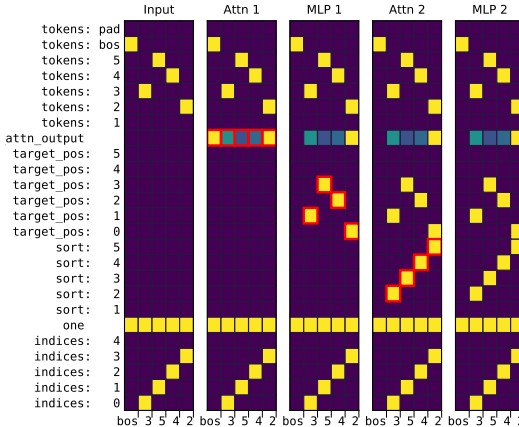

Figure 5: RASP program that sorts a sequence of numbers without duplicates. Attn 1 and MLP 1 implement the selector_width primitive (cf. Appendix D) which the program uses to compute the target position for each token. Attn 2 moves the tokens to the desired position, and MLP 2 is a no-op.

We are now ready to compile models with Tracr and walk through a few example programs.

## 4 Exploring Compiled Transformers

In this section, we walk through two example programs to illustrate how the compiled models work. While these models are not necessarily realistic, they represent configurations of weights that could, in principle, be learned. Examining these models can therefore be a powerful didactic tool for understanding how transformers perform complex computation, which we hope will expand our collective imagination for their inner workings.

We were able to compile RASP programs for all the tasks described in Weiss et al. (2021), though we had to modify a few programs to only use features supported by Tracr. Appendix G contains more examples.

### 4.1 Example 1: Counting tokens

Figure 2 shows our primary running example, the frac_prevs program, that computes the fraction of previous "x" tokens.

The compiled frac_prevs model has a 14-dimensional residual stream, but it uses 12 out of these for the input embeddings. The remaining two dimensions contain the main numerical variables used in the computation: is_x and frac_prevs (the output variable). The input embeddings have a few special dimensions. In particular, tokens:bos is the beginning of sequence token which we need to implement arbitrary attention patterns, and one is an input dimension that is always 1, used as a constant, e.g., to add a bias in MLP layers.

The compiled model uses one MLP layer and one attention head. However, because our model architecture always starts with an attention layer, the compiled model has four layers, with the first and last layers being no-ops. The first MLP layer computes the indicator variable is_x based on the input tokens. The following attention layer computes a causal attention pattern and uses it to compute the faction of previous "x" tokens.

### 4.2 Example 2: Sorting

As a second example, let us consider sorting a sequence of numbers. Figure 5 shows a sort_unique program that sorts a sequence of unique tokens.

The program computes uses a selector to select smaller tokens for each input token, and then uses the selector_width primitive in RASP to compute the target position for each token. selector_width counts the number of elements in each row of a selector that are 1, in this case the number of elements that are smaller than a given input token. selector_width can be implemented in terms of other RASP operations (Weiss et al., 2021). However, in Tracr we treat it as a primitive

that compiles directly to an attention and MLP layer (here Attn 1 and MLP 1). See Appendix D for more details. The model then uses a second attention layer to move each token to its target position.

Weiss et al. (2021) propose a sort program that can handle duplicates (cf. their Figure 13). However, that implementation uses a composite selector

```
select(tokens, tokens, <) or (
    select(key, key, ==) and select(indices, indices, <))
```

to treat duplicates, which is not currently supported by `Tracr`. In Appendix G, we provide an alternative implementation of `sort` that handles duplicates by adding a small multiple of `indices` to the keys and then applying `sort_unique`.

### 4.3  More examples

`Tracr` can compile a wide range of RASP programs. In Appendix G, we discuss several additional examples, leading up to a program to check balanced parentheses (*Dyck-n*). Our open-source `Tracr` implementation (https://github.com/google-deepmind/tracr) contains a library of even more example programs to compile.

## 5  Compressing Compiled Transformers

Superposition is an important phenomenon in large language models (see Section 2.3, Elhage et al. (2022b), and Scherlis et al. (2022)). But to the best of our knowledge, it has not yet been studied in detail for models with more than two layers or in transformer models executing multi-step algorithms. `Tracr` lets us examine these models, and we can force different levels of superposition by applying a gradient-descent-based compression algorithm.

In addition to helping us study superposition, compressed models could be more efficient and realistic. `Tracr` models can be sparse and inefficient because they reserve an orthogonal subspace of the residual stream for each s-op.

Here, we present two case studies of compressing compiled models using the `frac_prevs` and the `sort_unique` programs from Section 4. These sketch how `Tracr` can be practically useful in advancing interpretability research, while also giving a glimpse of how `Tracr` could be extended to produce more realistic models.

### 5.1  Gradient Descent Based Compression

We use a single linear projection $W \in \mathbb{R}^{D \times d}$ to compress the disentangled residual stream with size $D$ to a smaller space with dimension $d < D$. We modify the model to apply $W^T$ whenever it reads from and $W$ whenever it writes to the residual stream (see Figure 6). We freeze all other weights and train only $W$ using stochastic gradient descent (SGD). Since vanilla `Tracr` models are sparse and have orthogonal features, this process can be viewed as learning the projection from a "hypothetical disentangled model" to the "observed model" described by Elhage et al. (2022b).

We want the compressed model to minimise loss under the constraint that it implements the same computation as the original model. We train $W$ to minimise $\mathbb{E}_x[\mathcal{L}_{\text{out}}(W, x) + \mathcal{L}_{\text{layer}}(W, x)]$, where

$$\mathcal{L}_{\text{out}} = \text{loss}(f(x), \hat{f}_W(x)); \;\; \mathcal{L}_{\text{layer}} = \sum_{\text{layer } i} (h_i(x) - \hat{h}_{W,i}(x))^2$$

Here, $f(x)$ is the output of the compiled model for input $x$, $\hat{f}_W(x)$ is the output of the compressed model, and $h_i(x)$ and $\hat{h}_{W,i}(x)$ are the output vectors at layer $i$ of the respective models.

For categorical outputs, $\mathcal{L}_{\text{out}}$ is the softmax cross-entropy loss, whereas, for numerical outputs, it is the mean-squared error. $\mathcal{L}_{\text{layer}}$ is a regularization term that incentives the compressed model to match the per-layer outputs of the original model. To minimise this loss, the compressed model will have to approximate the computation of the original model but with a smaller residual stream. We give both loss terms equal weight, but we found other weighting factors give similar results in practice.

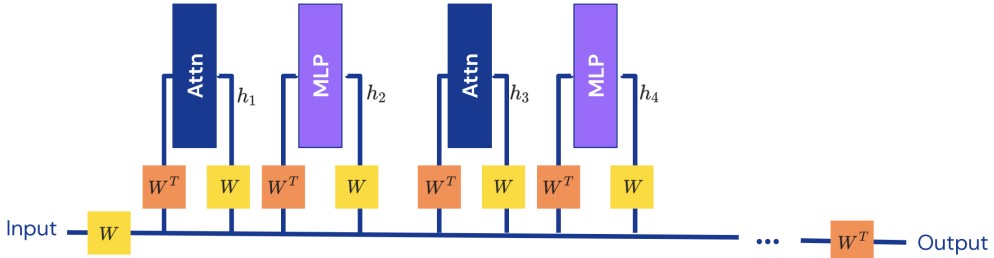

Figure 6: Training setup for compressing a compiled transformer model. At each layer, we use the same matrix $W \in \mathbb{R}^{D \times d}$ to embed the disentangled $D$-dimensional residual stream to $d \le D$ dimensions. We freeze the layer weights and only train $W$ to compress the model.

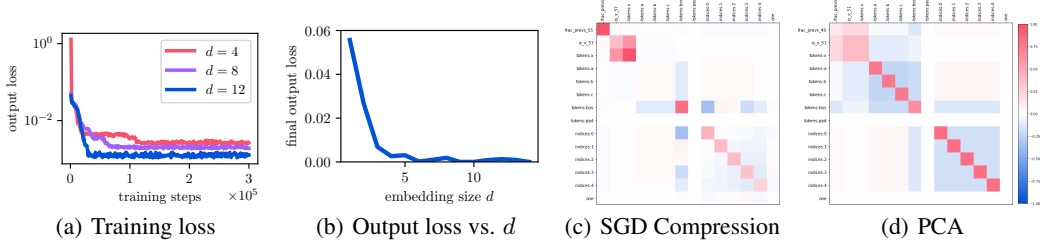

(a) Training loss      (b) Output loss vs. $d$      (c) SGD Compression      (d) PCA

Figure 7: Compressing the `frac_prevs` model Figure 2. (a) shows the loss during training for different embedding sizes $d$ and (b) shows the final loss for different embedding sizes $d$. After about $d = 6$ the compressed model solves the task essentially as well as the original compiled model which uses $D = 14$ dimensions. (c) shows $W^T W$ for the compression procedure described in Section 5 with $d = 8$ where $W$ is the learned compression matrix. As a comparison, (d) shows the same plot for applying PCA and retaining only the first 8 components. In contrast to PCA, our compression procedure produces a compression matrix $W$ that retains features necessary for the task (e.g., `is_x` and `frac_prevs`) and discards features that are unimportant (e.g., `tokens:a`).

We could set up this compression in other ways. For example, we could use a different projection at each layer, use different matrices for embedding and unembedding, or modify weights other than $W$ when compressing the model. These design choices come with a tradeoff between making the model more expressible and potentially more realistic and enforcing the ground truth computation. For simplicity, we use a shared $W$ for embedding/unembedding at every layer, and we already observe a rich structure in models compressed with this procedure.

Appendix E contains more details on the training setup, hyperparameters, and resources used.

## 5.2 What Does the Compression Learn?

As our first case study, Figure 7 shows the example model from Figure 2, that computes the fraction of token "x". By learning an embedding matrix $W$, we can reduce the residual dimension from $D = 14$ to $d = 6$ without hurting performance (cf Figure 7(b)). Once we reduce $d$ further, the model's performance starts to suffer.

To understand the compression better, we can study how $W$ embeds the original $D$ features in $d < D$ dimensions. We can only do this because we started with a compiled model with known features. Figure 7 shows $W^T W$ for compressing the model to $d = 8$. We can compare this to using principle component analysis (PCA) to compress the model. To interpret the results, we need to use our knowledge of the algorithm the model implements. The input `tokens:x` and the variables `is_x` and `frac_prevs` are crucial for computing the fraction of tokens that is "x", and we find that these variables mostly get separate dimensions in the compressed residual stream. The other input tokens stored in `tokens:a`, `tokens:b`, `tokens:c` are not necessary for solving the task, and so they are discarded in the compressed model. Other variables, such as the `indices` embeddings, are stored in non-orthogonal dimensions in the compressed space. This is consistent with existing findings on superposition as the `indices` embeddings are sparse and do not occur together (Elhage et al., 2022b).

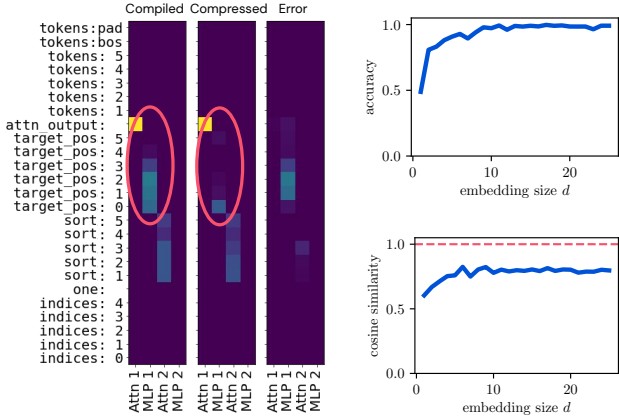

Figure 8: We compress `sort_unique` (Figure 5). The plots on the right show that the compressed model achieves nearly perfect accuracy, but the layer outputs of the compressed model are different from the original compiled model. The left plot shows the average layer outputs of the compiled model, the compressed model, and the squared difference. The compressed model seems to learn to use a different (numerical) encoding for the `target_pos` variable, which causes the discrepancy.

However, our results go beyond previous work on superposition. `Tracr` models often have multiple variables that depend on each other and encode shared information. For example, in `frac_prevs`, the `is_x` variable is an indicator that essentially contains the same information as the input dimension `tokens:x`.[4] In Figure 7, we see that the embeddings of `is_x` and `tokens:x` share part of the embedding space. Intuitively, this occurs because the variables encode similar information.

Future experiments could aim to further clarify the effect of shared information between variables on superposition. `Tracr` provides, for the first time, a setting to systematically study superposition in transformer models that implement nontrivial algorithms.

### 5.3 Do the Compressed Models Still Implement the Same Computation?

Even if the compressed models successfully achieve a low loss, we need to check if they implement the same computation as the compiled models, or else we would no longer know the ground truth mechanisms the models implement. To this end, we evaluate the average cosine similarity between the output at each layer of the two models. Values far from $1$ suggest the compressed model is structured differently from the base model.

We find that for some models the cosine similarity stays substantially below $1$ even as the compressed model gets close to $100\%$ in accuracy. For example, Figure 8 shows results from compressing the `sort_unique` model. Here, the compressed model achieves almost perfect accuracy on the task, but the average cosine similarity of the outputs at individual layers stays around $0.8$, far shy of $1$.

By inspecting the models' outputs at each layer, we can attribute the error to the `target_pos` variable. In the compiled model, `target_pos` is encoded as a one-hot vector. However, the compiled model only uses a single dimension. This suggests that the compressed model moves the tokens to the target position with a numerical encoding of the target position rather than a categorical encoding.

This difference in encodings shows that even with a fairly restrictive compression setup, compressed models may not stay faithful to the original RASP programs. This is both a setback for adding compression to the compiler—the compiler's annotations no longer serve as the exact ground truth—but also an opportunity. The ways neural networks solve algorithmic tasks regularly surprise researchers (Nanda et al., 2023). Studying such discrepancies could be a way to learn more about the ways NNs naturally represent certain computations without reverse-engineering entire models.

## 6 Related Work

There are many approaches to interpretability in machine learning (Carvalho et al., 2019), and in language models specifically (Danilevsky et al., 2020; Belinkov and Glass, 2019; Rogers et al., 2020). In this paper, we focus on interpretability in the sense of giving a faithful (Jacovi and Goldberg, 2020) and detailed account of the mechanisms learned by a model, sometimes called *mechanistic interpretability* (Olah, 2022) or *transparency* (Räukur et al., 2023).

---

[4]They are not exactly the same because `is_x` is only populated in a later layer.

Mechanistic interpretability has been used to reverse engineer circuits in state-of-the-art vision models (Cammarata et al., 2020), small transformer models trained on toy tasks (Olsson et al., 2022; Nanda et al., 2023), and medium-sized language models (Wang et al., 2023). Reverse-engineered circuits can be used as more realistic alternative to compiled models. However, they are labor-intensive to identify, and our knowledge of them can be incomplete or inaccurate even when they are analysed carefully. For example, Chan et al. (2022) show that the "induction head" hypothesis by Olsson et al. (2022) needs to be modified to adequately explain in-context learning performance even in small attention-only transformers.

While `Tracr` is based on RASP (Weiss et al., 2021), there are potential alternatives for constructing transformer models. Wei et al. (2022) and Akyürek et al. (2023) study more general computational models for transformers. Based on this line of work, Giannou et al. (2023) propose a Turing-complete model for constructing transformers, whereas RASP might have limited expressibility (Weiss et al., 2021; Merrill et al., 2022). However, the work by Giannou et al. (2023) is purely theoretical, and the practical cost-benefit trade-off between their approach and our RASP-based approach is unclear.

Evaluation is a perennial topic of debate in interpretability, and there is little consensus on the best approach (Lipton, 2018; Yang et al., 2019; Mohseni et al., 2021). We hope that compiled models contribute a new perspective to this discussion and can complement other evaluation methods.

Our approach is closest to prior work trying to create a ground truth for evaluating interpretability, via careful manipulation of the training mechanism and dataset. Yang and Kim (2019) and Adebayo et al. (2020) introduce label correlations to the background of images, and Zhou et al. (2022) use label reassignments to achieve a similar goal. However, these approaches focus on convolutional image classification models, and they can only modify part of a model to have a ground truth interpretation. `Tracr`, on the other hand, creates transformer models that implement fully human-readable code.

Since releasing an early version of our work, Conmy et al. (2023) successfully used `Tracr` to evaluate a method for automatically detecting circuits in transformer models, and Friedman et al. (2023) built on `Tracr` and studied *learning* transformer programs instead of manually writing them.

## 7   Discussion & Conclusion

We proposed to compile human-readable programs to neural network weights as a testbed for developing and evaluating interpretability tools. To this end, we introduced `Tracr` which compiles human-readable code to the weights of a transformer model.

**Applications.** Compiled transformer models can be broadly useful for accelerating interpretability research. We highlight four usecases that could be particularly useful. First, we can use `Tracr` to create test cases and ultimately benchmarks for interpretbility tools. This can help to confirm methods work as expected and surface potential failure modes. Second, we can measure our understanding of a model by manually replacing components of it with compiled components (similar to Nanda et al. (2023)). Over time, the research community could build a library of programs that represent our understanding of what neural networks learn. Third, we can use compiled models to isolate and study phenomena that occur in real neural networks. Our study of superposition in Section 5 demonstrates the benefits of studying an isolated phenomenon in a model we otherwise fully understand. Finally, compiled models can help us understand how transformers can implement certain algorithms and improve our ability to form concrete intuitions and hypotheses about models we want to interpret. Appendix A.1 discusses these applications in more detail.

**Limitations.** RASP and `Tracr` have important limitations in terms of expressivity, efficiency and realism compared to real transformer models. While many limitations can be overcome in future versions, some are fundamental to using compiled models. Clearly, we will likely never compile fully featured language models in `Tracr`. Therefore, we should interpret experiments conducted on compiled models carefully, and treat evaluations based on them as a minimum bar rather than a full validation of a technique. Appendix A.2 discusses these limitations in detail.

Despite these limitations, we think `Tracr` provides a promising new approach to studying transformers and to evaluating interpretability tools. The current approach to doing interpretability research is similar to trying to invent a microscope lens without ever being able to point it at familiar, well-understood shapes. `Tracr` enables researchers to point their interpretability methods at models they fully understand to calibrate, evaluate, and improve the methods.

## Acknowledgements

We thank Avraham Ruderman, Jackie Kay, Michela Paganini, Tom Lieberum, and Geoffrey Irving for valuable discussions, Victoria Krakovna and Marlene Staib for collaborating on early experiments with compiling RASP, and Chris Olah and Tristan Hume for feedback on an early draft of this paper. We thank the LessWrong user "Gurkenglas" for pointing out a mistake in an earlier draft of the way to implement selectors combined with `and` described in Appendix F.

## Author Contributions

VM proposed the initial idea for `Tracr` and wrote our RASP implementation. DL, VM, JK and MR designed and developed `Tracr`. DL designed, implemented, and ran the compression experiments in Section 5. MR wrote documentation and led the open-sourcing process. JK derived the theoretical results in Appendix F. TM and VM advised on research direction. DL, SF, and VM wrote the manuscript. DL led the project.

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

# A   Applications and Limitations of `Tracr`

We provide an open-source implementation of `Tracr` because we think it has many potential applications in interpretability research. In this section, we discuss applications we see for `Tracr` and compiled transformers more generally and reflect on the current limitations of `Tracr` and how they can be addressed.

## A.1   Applications of compiled models in interpretability research

Compilers like `Tracr` allow researchers to set up controlled experiments that test specific hypotheses about the computational structure of transformers. In this way, it acts as a laboratory for research in interpretability, enabling research that might otherwise be intractable.

**Understanding model phenomena and developing new techniques.** Compiled models can be used as a testbed for studying how learning affects circuits, and developing new approaches for interpreting transformer models. This is the approach we demonstrate in this work in section 5, where we successfully induce superposition in compressed `Tracr` models. Future work could analyse superposition in `Tracr` models, extending previous work in toy models (Elhage et al., 2022b; Scherlis et al., 2022). In particular, `Tracr` allows studying how the structure of computation implemented by a model affects which features will be stored in superposition. One goal for this line of research could be to predict how a specific `Tracr` model will be compressed, which features will be stored in superposition and how. A complementary approach is to try reversing the superposition induced by a compression procedure, e.g., using ideas from compressed sensing and dictionary learning (Donoho, 2006; Aharon et al., 2006).

**Test cases for interpretability tools.** Compiled models serve as a natural foundation for testing the faithfulness (Jacovi and Goldberg, 2020) of an explanation, and provide a way to falsify (Leavitt and Morcos, 2020) the explanations given by interpretability techniques that aim to describe the inner workings of models.

For instance, classifier probes (Belinkov, 2022; Bau et al., 2017) aim to determine the locations in the model where particular features are represented. A simple example of this approach is training linear classifiers using intermediate activations of a subject model as inputs. The performance of these classifiers at predicting some feature using activations from layer $L$ is then taken as a proxy for the extent to which the feature is represented at that layer. Applying this method and correctly interpreting its results is challenging (Belinkov, 2022). `Tracr`-compiled models provide an opportunity to see what these methods say about models whose representations we understand fully, contextualising their results on real models.

Ultimately, compiled models could be used to build libraries of test cases for interpretability tools, which could in turn enable quantitative evaluation metrics.

**Replacing model components.** Another way to evaluate our understanding of how a model works is to replace parts of the model with hand-coded components. For example, Nanda et al. (2023) test their understanding of how a transformer implements modular addition by replacing components of the model with their own idealised implementation and find that this can *increase* downstream performance, which is strong evidence that the proposed explanation is correct. While `Tracr` compiles an algorithm into a full transformer model, it could be adapted to only compile part of a model to replace part of a trained model. This could make it easier to evaluate our understanding of a large model.

**Building intuition for algorithms implementable by transformers.** Weiss et al. (2021) highlight that RASP can be used to gain intuition for how transformers might implement certain tasks. `Tracr` is a natural next step in this direction, spelling out the relationship between the program and a transformer implementing it in complete detail. We caution, however, that `Tracr` is but one approach to doing so, while real learned models could exhibit greater variety in their algorithms.

## A.2   Limitations of RASP and `Tracr`

RASP and `Tracr` are limited in terms of expressivity, efficiency and realism compared to real transformer models. Many of these limitations could be overcome in future versions of `Tracr`.

**Expressivity.** RASP is designed for algorithmic tasks that map an input sequence to a discrete output sequence. However, current language models usually map a sequence of input tokens to a probability distribution over the next token. Circuits in real models often consist of components that increase or decrease the probability of some tokens based on previous tokens (Wang et al., 2023). RASP, and hence `Tracr`, cannot model such "probabilistic" computation, but could potentially be extended to support it. RASP only uses binary attention patterns, which inherently limits the range of algorithms it can implement (Merrill et al., 2022). A way to extend RASP to support numeric attention patterns is discussed in Weiss et al. (2021).

**Efficiency.** `Tracr` models store all variables in orthogonal subspaces of the residual stream. Even if a variable is only used in part of the computation, `Tracr` reserves a subspace of the residual stream for it in all layers of the model. Real models use a more compressed representation and likely reuse dimensions for multiple features. Improved versions of the compression procedure discussed in Section 5 could address this limitation, as would using a constraint optimisation solver instead of a heuristic for layer allocation.

**Realism.** `Tracr` constructs layers from hand-coded parameter matrices. This is both unrealistic and inefficient, but could be addressed by learning the layers in isolation, then assembling them into a full model manually. Similarly, instead of manually splitting the $W_{QK}$ and $W_{OV}$ matrices, matrix factorisation could be used to get more efficient solutions. Also, `Tracr` models align their features with the computational basis. This is unrealistic, and makes the resulting models easy to interpret just by inspecting the residual stream activations. Rotating the basis of the compiled model is a straightforward way to address this if obfuscation is needed; compression would be an even more comprehensive approach.

While all of these issues could be overcome in a more sophisticated compiler, there are fundamental limitations on the role compiled models can play. Compiled models are an intermediate step between very simple toy models and real learned models. They help us understand ideas and methods, but results in compiled models do not necessarily generalise to real models. Compared with real models, compiled models will always be simpler. For example, we will likely never compile full-fledged language models. Compiled models will be more likely to be intepretable (e.g., the axis-aligned orthogonal residual stream bases in `Tracr`), and more likely to fit into existing paradigms for thinking about transformers. When using them to evaluate interpretability tools, we should be careful to make sure that the tools do not exploit this, treating such evaluations as a minimum bar rather than a full validation of a technique. Conversely, some methods might conceivably rely on features present in real models but not in compiled models.

# B   Modifications to RASP

**Disallow arbitrary selector combinations.** RASP allows boolean combinations of selectors; however, real transformers have no natural analogue. Combining selectors with different input variables is particularly problematic. For example, in RASP we can define a selector

```
select(A, B, ==) and select(C, D, ==)
```

using four s-ops `A`, `B`, `C` and `D`. However, a real attention pattern only has two input vector spaces. There is no straightforward and efficient construction for representing arbitrary compositions of selectors (appendix F). Because of this, we restrict RASP to selectors with only two input variables. In practice, this limitation seems not severe. In particular, we could implement programs to solve all tasks described by Weiss et al. (2021).

If a composite selector cannot be avoided, one can always refactor it into an atomic selector by first using s-ops to create a product spaces over the inputs. In the example above, we'd construct two s-ops whose values are pairs over values of `A`, `B` and `C`, `D` respectively. Then, we could construct an atomic selector operating on these composite s-ops:

```
select(
    product(A, B),
    product(C, D),
    lambda (a,b), (c,d): a==c and b==d
)
```

While this refactoring can be done mechanically, and would naturally generalise to arbitrary selector combinations, we chose not to include it in our compiler implementation for two reasons. First, without compression it is inefficient: the s-op dimensions scale as a product of the input s-op dimensions. Second, doing this automatically would break the 1-1 correspondence between selectors in RASP and attention heads in the compiled model: the compound s-ops require MLP blocks.

**Encoding annotations.** A compiled model needs to pass information between layers. In a transformer, it is natural to do this in the residual stream (Elhage et al., 2021). However, our compiler must decide how to represent information in the residual stream. For simplicity, we only use categorical and numerical encodings. We encode categorical variables as one-hot vectors in a dedicated subspace of the residual stream. We encode numerical variables as the magnitude of a dedicated one-dimensional subspace of the residual stream. We require each s-op to be either categorical or numerical and augment RASP to annotate s-ops with the desired encoding. S-ops are categorical by default.

Even when both categorical and numerical encodings are possible for the same information, categorical encoding generally uses more dimensions and often requires an extra decoding step. However, some aggregate operations only work with one type of encoding. For instance, aggregation with a mean across token positions is natural for numerical encodings but not categorical ones.

**Beginning of sequence token.** Transformers often assume any input sequence starts with a dedicated "beginning of sequence" token (BOS). We make the BOS token mandatory in RASP because it is crucial when implementing arbitrary attention patterns. In particular, RASP allows selectors that can produce all-zero rows; this is convenient when programming in RASP, but the softmax makes this behaviour impossible in a real attention head. In these situations, we use the BOS token as a "default" position to attend to: it is attended to iff no other token is. This allows the non-BOS part of the sequence to emulate the intended RASP behaviour. In our case, this choice comes from practical considerations; but, interestingly, real models sometimes show similar behaviour (e.g., see Elhage et al., 2021).

## C  Reading Model Output Figures

In the main paper and Appendix G, we show figures of a forward pass in a compiled model. We found that these figures can be confusing to read at first, especially as the compiled models get bigger. This section serves as a reference for how to interpret these figures.

As an example, let us walk through the figure for the `frac_prevs` model from Figure 2:

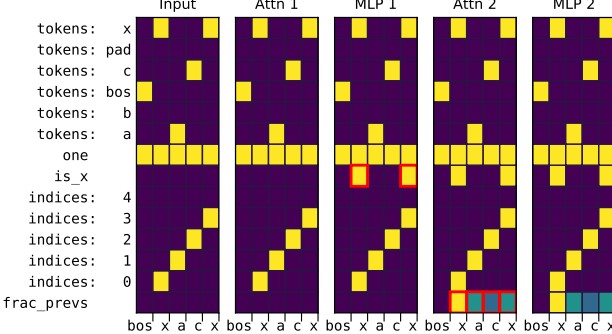

The figure has 5 panels, each of which shows the content of the residual stream after the corresponding layer in the model. This allows us to follow what the model does step-by-step. The residual stream has size `[sequence length x dimensionality]`, therefore we visualize it as a 2-dimensional heatmap. In this example, we have a size of $5 \times 14$ including the BOS token position.

We show a forward pass for a specific input sequence `[bos, x, a, c, x]`. On the x-axis of each panel we label the token positions with the corresponding input token. The y-axis of the plot contains the dimensions of the residual stream. Thanks to our knowledge of the program the model implements, we can label each dimension according to what it encodes. Dimensions starting with 'tokens' contain the (categorical) input embeddings. Dimensions starting with 'indices' contain the (categorical) position embeddings. Labels that contain a ':' are dimensions that correspond to a categorical ('one-hot') enoding and the value after the ':' is the value encoded in this dimension.

Labels without a ':' mean that this dimension encodes a numerical value. In each of the four panels we show the full residual stream content as a heatmap. Entries that were changed by the layer corresponding to the panel are highlighted with a red border.

Let's go through the plot step by step and map it to the code we used to compile this model:

```
is_x = (tokens == "x")
prevs = select(indices, indices, <=)
frac_prevs = aggregate(prevs, is_x)
```

In the leftmost panel we see the residual stream after the input embedding layer. It contains the categorical encoding of the `tokens` and the categorical encoding of the `indices`. For example at the token position of the 'a' token, the dimension `tokens:a` contains 1, and the dimension `indices:1` contains 1. The auxiliary dimension `one` contains 1 at every token position.

The first attention layer is a no-op, so the residual stream afterwards (shown in the second panel) is the same as before. No entry is highlighted.

MLP 1 computes the first line of the rasp program `is_x = (tokens == "x")`. It writes the result into a numerical dimension in the residual stream labelled with `is_x`. For this concrete sequence the layer writes a 1 into both token positions that contain the token 'x' in the input.

Attn 2 computes the select-aggregate operations in lines 2 and 3 of the RASP program. It computes the fraction of previous 'x' tokens, and writes the result into a single dimension labelled `frac_prevs`. It writes values between 0 and 1 in all token positions except for the BOS token position. For this example the result will be `[1, 1/2, 1/3, 1/2]`.

The final MLP 2 layer is a no-op again and does not change anything in the residual stream. The output unembedding layer will then read the result from the `frac_prevs` dimension.

## D  `Tracr` Implementation Details

This section highlights a few more implementation details of `Tracr`. We describe how we construct MLP and attention blocks, how we implement the selector width primitive, and how we extend RASP and `Tracr` to use causal attention. For the full implementation and documentation, refer to the code repository at https://github.com/google-deepmind/tracr.

### D.1  MLP and Attention Blocks

For MLP layers, we distinguish between `Map` operations with a single input and output and `SequenceMap` operations with two inputs and one output. We can recursively represent functions with more than two inputs using `SequenceMaps`.

We translate `Maps` with categorical inputs and outputs to MLPs that act as a lookup table. `SequenceMaps` with categorical inputs and outputs become MLPs where the first layer maps to an encoding of all pairs of inputs and the second layer acts as a lookup table.

For numerical inputs and outputs, we explicitly construct MLP layers as universal function approximators. In these MLPs, the first layer discretises the input, and the second layer maps each discrete bucket to a corresponding output value. We know which input/output values can occur, so we can choose the discretisation around these known input values to minimise the approximation error.

We now turn our attention to the attention blocks, which we construct from RASP selectors.

We first construct the $\tilde{W}_{QK}$ matrix to implement the desired attention pattern in the attention logits. We will refer to this as the *direct attention matrix*. This matrix has low rank, with its row space being the part of the residual stream where the query s-op is stored, and the column space being where the key s-op is stored. We adjust the direct attention matrix matrix by adding a rank-one update $W_{BOS} = \beta_{BOS} x_{\texttt{one}} x_{\texttt{tokens:bos}}^{\mathsf{T}}$ with $\beta_{BOS} = 1$ or $\beta_{BOS} = \frac{1}{2}$, to ensure that the BOS token is attended to either always, or whenever no other token is. ($x_{\texttt{one}}$ and $x_{\texttt{tokens:bos}}$ here are unit vectors for the special embedding dimensions introduced in Section 4.) We then scale up the matrix by an inverse-temperature parameter $T^{-1}$ (100 by default), getting $W_{QK} = T^{-1}(\tilde{W}_{QK} + W_{BOS})$. As a result,

the attention weights $A_{ij} = \text{softmax}\left(\mathbf{q}_i^\intercal W_{QK}\vec{\mathbf{k}}\right)_j = \exp(\mathbf{q}_i^\intercal W_{QK}\mathbf{k}_j)/\sum_{j'}\exp(\mathbf{q}_i^\intercal W_{QK}\mathbf{k}_{j'})$ are very close to $1/\#\{\text{selected tokens}\}$ on selected tokens and 0 elsewhere.

The $W_{OV}$ matrix maps the value input to the corresponding output dimensions. Attention layers only support categorical key and query inputs. The value inputs can be numerical or categorical. We can only use categorical values if the head never attends to more than one token.

## D.2 Selector Width Primitive

RASP provides the selector width primitive, which counts the number of 1s in each row of a selector. It provides an alternative to `aggregate` for processing selectors.

Weiss et al. (2021) provide a selector width implementation in pure RASP, making it not necessarily a language primitive. However, the most efficient implementation uses the BOS token, which exists `Tracr` but is not exposed to the RASP program.

Therefore, `Tracr` translates selector width directly into an efficient implementation in `Craft` consisting of an attention layer and an MLP layer. The attention layer implements an attention pattern that matches the selector to compute the width of. It uses the BOS token as value input, resulting in the attention head computing $x = 1/(1 + w)$ where $w$ is the desired selector width output. The next MLP layer then computes $w = 1/x - 1$ and cleans the BOS token position.

## D.3 Causal Attention

Most transformer models used in practice use causal attention, i.e., they apply a mask to the attention patterns that allows the model to attend only to previous tokens. This allows training the models autoregressively. However, RASP assumes non-causal (i.e. bidirectional) attention by default. While all models discussed in the main paper use non-causal attention, `Tracr` also supports causal attention.

To enable this, we extend RASP to support causal attention via a flag set during evaluation. To evaluate a RASP program in the causal evaluation mode, we apply a causal mask to the output of each selector. Causal evaluation changes the semantics of some RASP operations, and, in general, it is necessary to adapt RASP programs to function with causal attention. For example, the `frac_prevs` program no longer needs to compute a causal mask manually. However, for example, the `length` implementation by Weiss et al. (2021) no longer correctly computes the length of a sequence because it requires attending to future tokens.

Similarly, `Tracr` has a flag to enable causal compilation. Most of the compilation process does not change, and we only need to ensure to compile selectors to causal attention heads.

# E   Compression Training Details

We implemented the compression described in Section 5 in Jax on top of the Haiku transformer implementation that comes with `Tracr`. We train $W$ using the AdamW optimizer (implemented in Optax) with a weight decay factor of 0.1, and parameters $\beta_1 = 0.9, \beta_2 = 0.99$. We train for $3 \times 10^5$ steps with a batch size of 256. We decay the learning rate linearly from $10^{-3}$ to $10^{-6}$ over the first half of training. Each compression run requires between 1 and 4 hours of run time on two CPU cores (depending on the size of the model to compress).

# F   Theoretical Results on Combining Attention Heads

The RASP language permits combining arbitrary selectors elementwise using boolean operators, such as `and`, `or`, and `not`. It is not immediately obvious what operators can be implemented given the way we encode selectors as attention matrices $W_{QK}$, as described in Appendix D.1.

First, let's consider `not` operator for a selector `select(query, key, pred)` with given direct attention matrix $\tilde{W}_{QK}$. One way to implement `not select(query, key, pred)` is to note that it's equivalent to `select(query, key, not pred)`. Another is to use a transformed direct attention matrix $\tilde{W}_{QK}^{\text{not}} = -\tilde{W}_{QK}$, alongside a $\beta_{BOS}^{\text{not}}$ that's 0 or $-\frac{1}{2}$.

Next, let's consider the `and` operator on two selectors `select(query_a, key_a, pred_a)` and `select(query_b, key_b, pred_b)` whose direct attention matrices $\tilde{W}_{QK}^A, W_{QK}^B$ are given, and produce 0-1 attention logits. We can observe that taking $\tilde{W}_{QK}^{\text{and}} = \tilde{W}_{QK}^A + \tilde{W}_{QK}^B$ results in attention logits taking value 2 when both selectors are active, and at most 1 otherwise; so by the same procedure in Appendix D.1, with $\beta_{BOS}^{\text{and}}$ taking value $\frac{3}{2}$ or 2, we can construct $W_{QK}^{\text{and}} = T^{-1}(\tilde{W}_{QK}^{\text{and}} + W_{BOS}^{\text{and}})$ that produces the desired attention pattern in the post-softmax attention weights.

We can compose these constructions, negating the two given selectors before combining them with `and`, to get `nor`, with $\tilde{W}_{QK}^{\text{nor}} = -\tilde{W}_{QK}^A - \tilde{W}_{QK}^B$ and $\beta_{BOS}^{\text{and}}$ taking value $-\frac{1}{2}$ or 0, resulting in an implementation of `select(query_a, key_a, pred_a)` `nor` `select(query_b, key_b, pred_b)`.

So far these are fairly natural constructions – the boolean operators `not` and `and` can be used to construct all other possible boolean operators, so we might expect that indeed all combinations of selectors via boolean operators can be compiled to transformer weights this way.

Alas, it is not so. Unlike the implementation of `not`, the implementations of `and` and `nor` above did not result in a direct attention matrix that produces the correct pattern (potentially shifted by a constant) in the attention logits, but rather only in the attention weights after temperature-adjusted softmax, meaning they cannot be composed further to produce arbitrary logical statements.

If we were to try to implement `or`, the easiest way would be to negate the `nor` by composing the transformations – but the resulting $\tilde{W}_{QK}^{\text{or}} = -(-\tilde{W}_{QK}^A - \tilde{W}_{QK}^B)$ is actually the same direct attention matrix we used for `and`. This produces attention logit 1 or 2 where the selectors' `or` is active, and 0 where it isn't. However, the temperature adjustment with $T^{-1} \gg 1$ that forces the attention to be near-zero where neither selector is active will then also do the same thing when only one selector is active, so the attention weights will be different between tokens where both selectors are active versus only one selector.

In fact, this obstruction to implementing `or` can be generalized, as follows.

**Lemma F.1.** *Consider two selectors `select(query_A, key_A, pred_A)` and `select(query_B, key_B, pred_B)`, with direct attention matrices $\tilde{W}_{QK}^A$ and $\tilde{W}_{QK}^B$. For ease of analysis, let's suppose `query_A`, `key_A`, `query_B`, and `key_B` are stored in separate, orthogonal subspaces $Q_A$, $K_A$, $Q_B$, $K_B$.*

*Now suppose there exists an attention matrix $\tilde{W}_{QK}^{\text{or}}$, with row space contained in $Q_A + Q_B$ and column space contained in $R_A + R_B$, that, after adjustment by some BOS logit offset $\beta_{BOS}^{\text{or}}$ and some temperature $T \to 0$, produces attention weights converging to the normalized selector weights for `select(query_A, key_A, pred_A)` `or` `select(query_B, key_B, pred_B)`. Then, these selectors are not generic – they satisfy some very limiting constraints about their predicates.*

*Proof.* Let's begin by assuming the second selector, $B$, is not constant, selecting some tokens and not-selecting other tokens. This implies the existence of basis vectors $\mathbf{q}_B^0, \mathbf{q}_B^1 \in Q_B$ and $\mathbf{k}_B^0, \mathbf{k}_B^1 \in K_B$ such that ${\mathbf{q}_B^0}^\intercal \tilde{W}_{QK}^B \mathbf{k}_B^0 = 0$ and ${\mathbf{q}_B^1}^\intercal \tilde{W}_{QK}^B \mathbf{k}_B^1 = 1$. Holding these constant, consider some basis vectors $\mathbf{q}_A \in Q_A$ and $\mathbf{k}_A, \mathbf{k}_A' \in K_A$. Then, for query vector $\mathbf{q}_A + \mathbf{q}_B^1$, all tokens with key vector $\mathbf{k}_A + \mathbf{k}_B^1$ or $\mathbf{k}_A' + \mathbf{k}_B^1$ must be selected, which means they must have equal attention logits. Therefore, $(\mathbf{q}_A + \mathbf{q}_B^1)^\intercal \tilde{W}_{QK}^{\text{or}}(\mathbf{k}_A + \mathbf{k}_B^1) = (\mathbf{q}_A + \mathbf{q}_B^1)^\intercal \tilde{W}_{QK}^{\text{or}}(\mathbf{k}_A' + \mathbf{k}_B^1)$, so $\mathbf{q}_A^\intercal \tilde{W}_{QK}^{\text{or}} \mathbf{k}_A = \mathbf{q}_A^\intercal \tilde{W}_{QK}^{\text{or}} \mathbf{k}_A'$.

Now, consider $\mathbf{k} = \mathbf{k}_A + \mathbf{k}_B^0$, $\mathbf{k}' = \mathbf{k}_A' + \mathbf{k}_B^0$, $\mathbf{q} = \mathbf{q}_A + \mathbf{q}_B^0$, and, for some basis vector $\mathbf{q}_A' \in Q_A$, let $\mathbf{q}' = \mathbf{q}_A' + \mathbf{q}_B^0$. We have logit differences $\mathbf{q}^\intercal \tilde{W}_{QK}^{\text{or}} \mathbf{k}' - \mathbf{q}^\intercal \tilde{W}_{QK}^{\text{or}} \mathbf{k} = {\mathbf{q}_B^0}^\intercal \tilde{W}_{QK}^{\text{or}}(\mathbf{k}' - \mathbf{k}) = \mathbf{q}'^\intercal \tilde{W}_{QK}^{\text{or}} \mathbf{k}' - \mathbf{q}'^\intercal \tilde{W}_{QK}^{\text{or}} \mathbf{k}$. Therefore, among tokens where `key_B` has vector $\mathbf{k}_B^0$ (let's call these $\mathbf{k}_B^0$-tokens), the tokens that have highest logit for query vector $\mathbf{q}'$ are the same as those for query vector $\mathbf{q}$. However, the selected tokens among the $\mathbf{k}_B^0$-tokens are either none of them, or exactly those with the highest logit (which depends on `key_A`). Because of the definition of `or`, $\mathbf{k}_B^0$-tokens are selected exactly if `select(query_A, key_A, pred_A)` would select them.

Putting the above observations together, it follows that for `query_A` vectors $\mathbf{q}_A$ and $\mathbf{q}_A'$, `pred_A` will either select no keys for one of them, or will select the same keys for both of them. In other words, `pred_A` must be rewritable in the form `query_pred_A(query_A)` `and` `key_pred_A`

(key_A). Equivalently, `pred_A`'s matrix has rank 1; we can say in short that `pred_A` is a rank-1 predicate, or that `select(query_A, key_A, pred_A)` is a rank-1 selector.

If we suppose our initial assumption to be false, then `pred_B` is constant, and can thus be just as well rewritten to be a predicate of `query_A` and `key_A`; then, it is easy to derive the necessary $\tilde{W}_{QK}^{or}$ from `select(query_A, key_A, pred_A or pred_B)`.

We can repeat the argument interchanging the selectors, to conclude that either the operation is trivial (because one predicate is constant), or both selectors must be rank-1. □

The above conclusion may be averted in the case that we have a priori information that certain values of $\mathbf{q}_A, \mathbf{k}_A, \mathbf{q}_B, \mathbf{k}_B$ cannot co-occur, or if some of the input s-ops are shared. We leave exploring that, as well as whether `or` can be implemented in the case of rank-1 predicates, to future work.

A notable special case of the above is the case where `query_A` and `query_B` compute the same s-op, and `key_A` and `key_B` also compute the same s-op. (They may be the same s-op, or redundant copies.) Then simple rewriting is possible, similarly to the `or` case explained earlier. For example:

```
simplifiable_selector = select(tokens, indices, <=) or select(tokens, "a", ==)
simplified_selector = select(tokens, indices, q <= k or q == "a")
```

A similar strategy of matching s-ops can be used to circumvent the lemma and straightforwardly implement operators like `or`, by constructing combined s-ops `query_both` and `key_both` with output types representing all pairs of queries and keys of the two selectors. These s-ops may be computed by the preceding MLP – however, the encodings occupy dimensionality multiplicative in the sizes of the constituent s-op output types, which is an impediment to scaling these circuits very far.

Due to the composability limitations of each approach considered, we did not implement boolean operators acting on selectors, apart from simple cases where the query and key s-ops agree.

## G More Compiled Models

Here, we present a few additional RASP programs and the compiled `Tracr` models.

Figure 9 shows and extended `sort` program. It works similarly to the `sort_unique` program in Figure 5, but sorts any sequence of values by a sequence of keys and can handle duplicates occurring in the keys.

Figure 10 shows the `pair_balance` program, which computes the difference in the fraction of open and closed parenthesis tokens. We can now use it as a subroutine for the `dyck-n` program, which checks if a sequence of $n$ different types of parentheses is balanced:

**Input**: `pairs`

```
1  # Compute running balance of each type of parenthesis
2  balances = [pair_balance(pair) for pair in pairs]
3
4  # If balances were negative anywhere -> parentheses not
       balanced
5  any_negative = balances[0] < 0
6  for balance in balances[1:]:
7      any_negative = any_negative or (balance < 0)
8
9  select_all = select(1, 1, ==)
10 has_neg = aggregate(select_all, any_negative)
11
12 # If all balances are 0 at the end -> closed all parentheses
13 all_zero = balances[0] == 0
14 for balance in balances[1:]:
```

```
15      all_zero = all_zero and (balance == 0)
16
17  select_last = select(indices, length - 1, ==)
18  last_zero = aggregate(select_last, all_zero)
19
20  dyck_n = (last_zero and not has_neg)
```

Figure 11 shows the compiled dyck-2 model for pairs = ("()", "{}").

**Input**: keys, vals, min_key, context_length

```
1  keys = (keys + indices + min_key) / context_length
2  smaller = select(keys, keys, <=)
3  target_pos = selector_width(smaller)
4  sel_sort = select(target_pos, indices, ==)
5  sort = aggregate(sel_sort, vals)
```

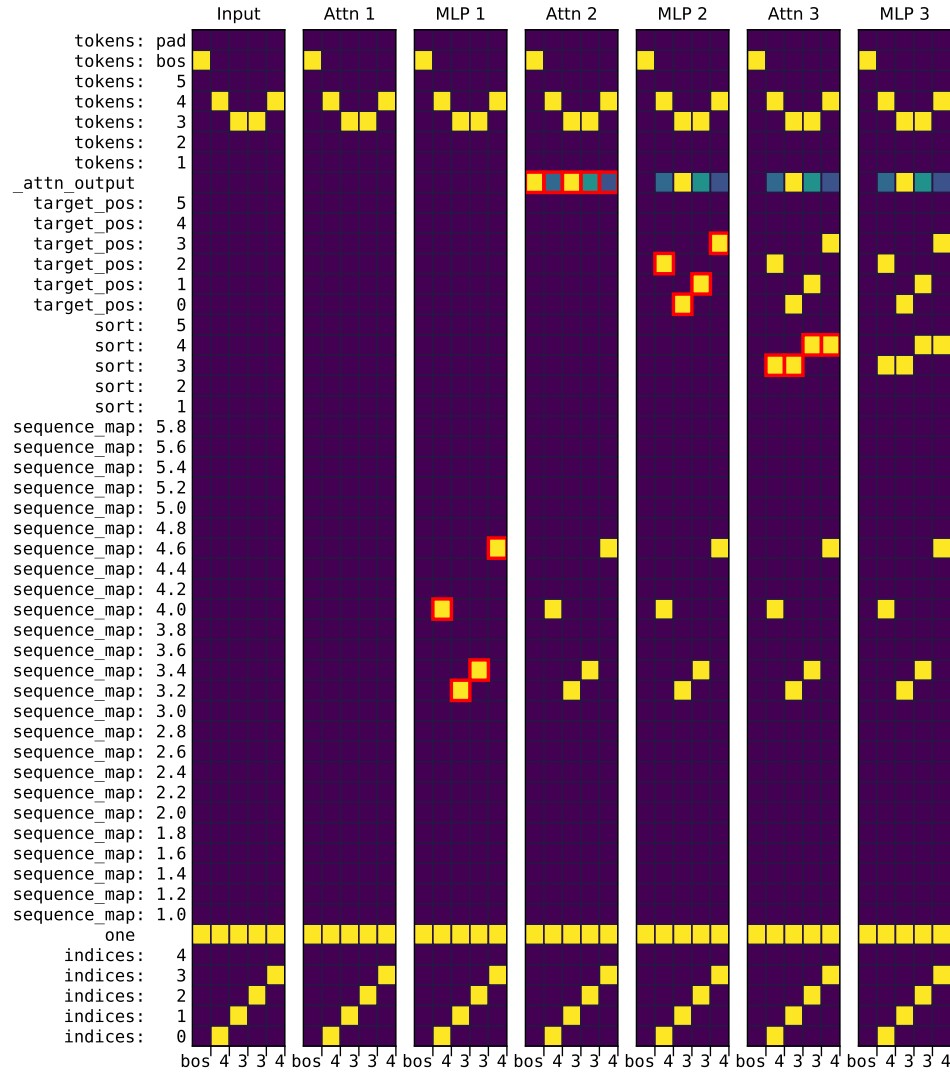

Figure 9: Compiled `sort` program. Attn 1 is a no-op, MLP 1 adds a small multiple of `indices` to the keys, and the rest of the model essentially implements `sort_unique`.

**Input**: `open_token`, `close_token`

```
1  bools_open = (tokens == open_token)
2  opens = frac_prevs(bools_open)
3  bools_close = (tokens == close_token)
4  closes = frac_prevs(bools_close)
5  pair_balance = opens - closes
```

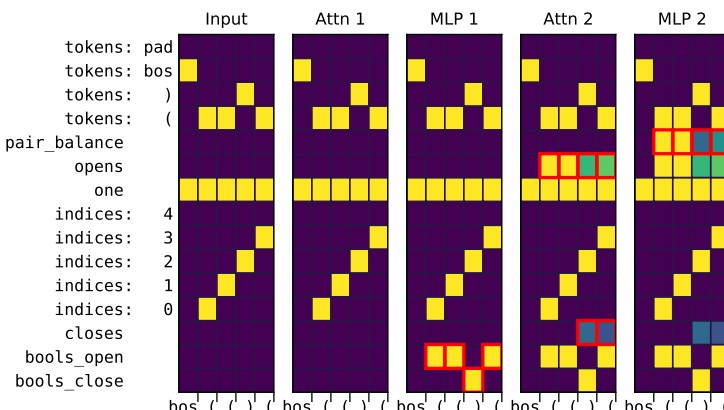

Figure 10: RASP program that uses `frac_prevs` as a subroutine to compute the fraction of open and closed parenthesis tokens and computes the difference. The compiled model uses `open_token` = "(" and `close_token` = ")". Note that the compiled model has the same number of layers as the single `frac_prevs` model in Figure 2. Attn 1 is still a no-op, MLP 1 and Attn 2 compute both calls to `frac_prevs` in parallel, and MLP 2 computes the final result.

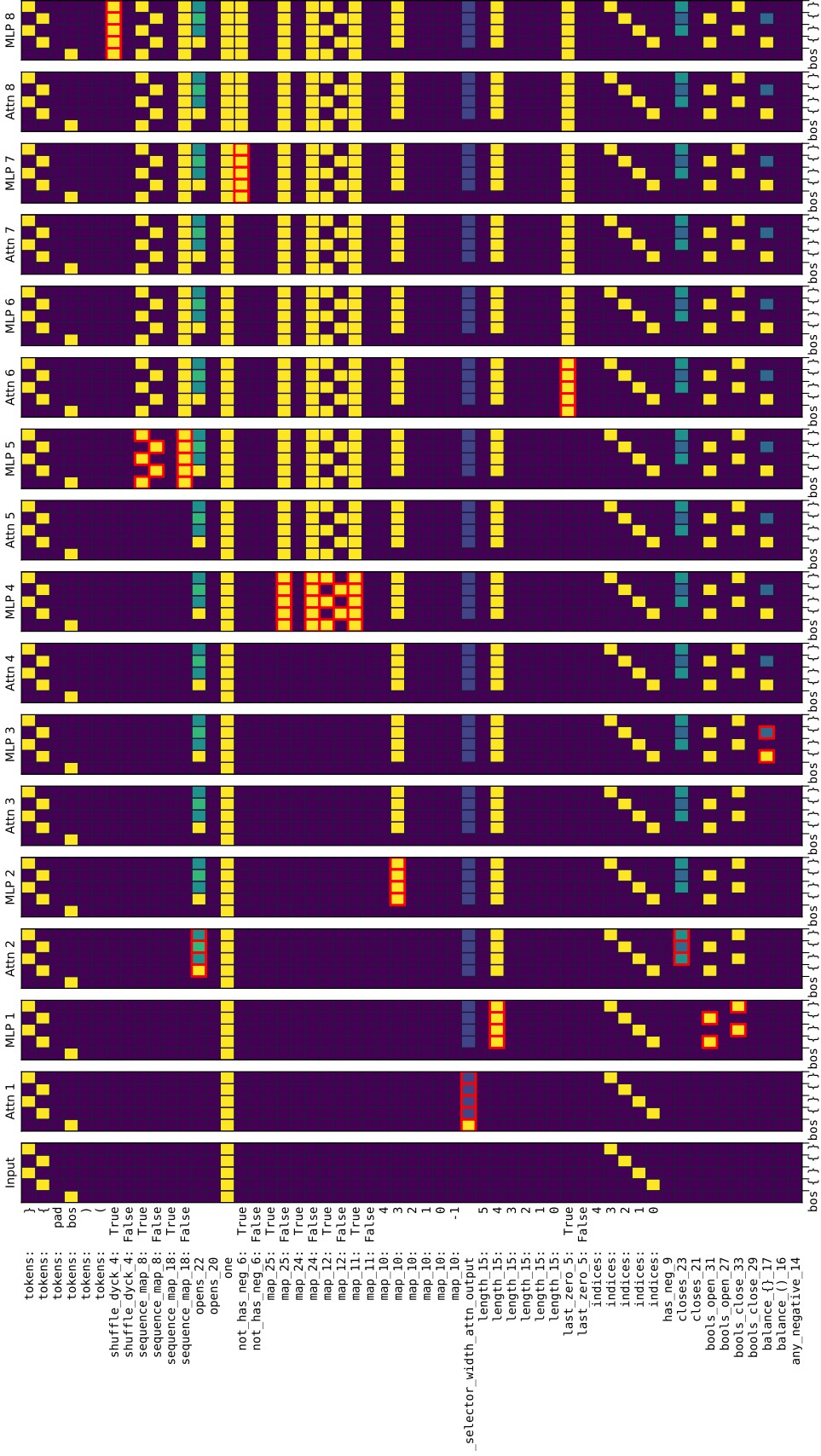

Figure 11: Compiled dyck-2 program for pairs = ("()", "{}").

