# OpenReview forum: "Tracr: Compiled Transformers as a Laboratory for Interpretability"
_NeurIPS.cc/2023/Conference — NeurIPS 2023 spotlight_

### Official Review · Reviewer_ACvV · 2023-06-08

**Soundness:** 3 good
**Presentation:** 4 excellent
**Contribution:** 3 good
**Rating:** 6
**Confidence:** 5

**Summary:**

This submission builds on the RASP language proposed by Weiss et al. It proposes a way to compile RASP program into real Transformer weights. It also comes with a case study showing the use case of Tracr to study a phenomenon called superposition which is widely known in Mechanistic Interpretability field. It's claimed by the authors that their Tracr can serve as a ground-truth for interpretability research.

**Strengths:**

This piece of construction result is of evident significance to the community, considering Transformer is the dominant architecture these days. It furthers our understandings theoretically how Transformer can be implementing various algorithms inside of it. I hope this can inspire us to discuss what can and what **cannot** be expressed in a specified architecture.

**Weaknesses:**

+ On the first use case: it is studying a phenomenon not even well defined in the literature, called superposition. Wish to see more rigorous framing of what is really studied, e.g., compressed sensing. The results in Section 5 isn't impressive to me since it isn't an organic combination of compressing and Tracr.

+ On the second use case: It feels to me an overclaiming that Tracr could serve as a _ground-truth_ for evaluating interpretability methods. This is an understandable imagination but such idealism isn't correct. An interpretability algorithm can discover very different underlining algorithms and if the discovered is different from Tracr compilation; only the Tracr is to be challenged.

+ Summary up, I acknowledge this is an important theoretical results but framing it as interpretability or anything related to real Transformer behaviour sounds far-fetched.

**Questions:**

+ On Line 209. The choice of $W^\top$ in Elhage et. al. to recover the compressed feature actually make more sense than in the case of this submission. They are forcing $W$ to be orthonormal so that $WW^\top=I$ stands, which is believable if it happens since their loss function is encouraging that. In Tracr's case, I cannot see why the residual added into the stream is supposed to be the same as the input.

**Limitations:**

+ This is indeed a good pedagogical tool for Transformer architecture. However, it's worth highlighting in the paper that LayerNorm or any similar operations, RMS Norm are left out, so to not mislead readers.

---

> ### Author Rebuttal · Authors · 2023-08-07
>
> > On the second use case: It feels to me an overclaiming that Tracr could serve as a ground-truth for evaluating interpretability methods. This is an understandable imagination but such idealism isn't correct. An interpretability algorithm can discover very different underlining algorithms and if the discovered is different from Tracr compilation; only the Tracr is to be challenged.
>
> This might be a misunderstanding of what we mean by ground-truth. We do not mean that if we can compile an algorithm in Tracr, this algorithm is necessarily the correct interpretation for a learned transformer model completing the same task. Say we compile a model using Tracr to sort a sequence of numbers and train a transformer to do the same task. You are correct that the trained model might use a different algorithm from the compiled model. The advantage of the compiled model is that if we run an interpretability method on it, we know exactly the correct interpretation for _that compiled model_ – the RASP program serves as a ground-truth. We can then see how that ground truth is reflected in the interpretability method.
>
>
> > On Line 209. The choice of in Elhage et. al. to recover the compressed feature actually make more sense than in the case of this submission. They are forcing to be orthonormal so that stands, which is believable if it happens since their loss function is encouraging that. In Tracr's case, I cannot see why the residual added into the stream is supposed to be the same as the input.
>
> You are right; there is no principled reason why the model needs to use the same embedding at each point in the model. We choose the same embedding matrix throughout the model primarily for simplicity, but also because this is how uncompressed Tracr models are structured. As mentioned in response to Reviewer 767m, we found our compression procedure to be a particularly good tradeoff between being simple and producing more efficient and natural models. However, it is worth noting that recent empirical evidence from real transformers suggests that the input embedding matrix is still a meaningful interpretation of the residual stream at other points of the model (e.g., see [1]).
>
>
> > However, it's worth highlighting in the paper that LayerNorm or any similar operations, RMS Norm are left out, so to not mislead readers.
>
> Thanks, we will make sure to highlight this point.
>
> **References**
>
> [1] Guy Dar, Mor Geva, Ankit Gupta, and Jonathan Berant. 2023. Analyzing Transformers in Embedding Space. In Proceedings of the 61st Annual Meeting of the Association for Computational Linguistics (Volume 1: Long Papers).

---

> > ### Comment · Reviewer_ACvV · 2023-08-10
> > **Thank the authors for the response**
> >
> > I appreciate the reply. Here is a little concern I still wish to hear your thoughts on: you mentioned "uncompressed Tracr models are structured", then how do you think of the possibility that there is a unbridgeable gap between real Transformer behavior and the Tracr (post-compression) compiled Transformer. If we apply MI techniques to Tracr Transformer and successfully rediscover the Tracr algorithm, does it mean anything that this MI technique can generalize to real Transformer? I.e., should we therefore trust any conclusion this MI get on real Transformer?
> >
> > To be clear, by no means I'm against the acceptation or any reward of this paper. Just curious about how you think about this.

---

> > > ### Author Response · Authors · 2023-08-12
> > >
> > > We agree that because of structural differences between compiled and real transformers, we should be careful when interpreting results obtained using Tracr models. At present, as we mention in the paper, we would suggest seeing them as a minimum bar for methods to pass rather than a full validation.
> > >
> > > If we find MI methods can pass this bar by unfairly exploiting structure specific to Tracr models that we don't expect in real models, future work can add additional layers of obfuscation and complexity to the compiler which would make it a more rigorous test.
> > >
> > > However, we think that even with more advanced compilers, compiled models are always going to be to some extent artificial, so such evaluations will never fully replace evaluations on real models. We see evaluations on compiled models and on real models as complementary approaches with different strengths and weaknesses.

---

> > > > ### Comment · Reviewer_ACvV · 2023-08-13
> > > > **Thank the authors for the response**
> > > >
> > > > Thank you for your reply!
> > > > Looking forward to seeing how this line of work grow in the future!

---

### Official Review · Reviewer_XH5L · 2023-07-03

**Soundness:** 3 good
**Presentation:** 3 good
**Contribution:** 3 good
**Rating:** 8
**Confidence:** 4

**Summary:**

The paper presents Tracr, which is a compiler from RASP programs (a language designed to showcase a possible computational model used by Transformers) to Transformer architectures and weights. The paper first details the operation of the compiler, then discusses examples of several simple RASP programs and their respective Tracr-generated Transformer models. Finally, the paper experiments with compressing the dimensionality of these models, exploring an emerging hypothesis about superposition of feature representation in Transformers.

**Strengths:**

* The paper addresses a significant gap in prior work, which is that while RASP purports to encode the computational model for Transformers, RASP programs are not directly compilable to Transformer weights. This is an active and interesting area of research, and providing such a compiler is likely to significantly impact work in this domain.
* Overall, the paper is very clearly written
* The examples given (Section 4) strongly demonstrate the success of the technique, and broadly help to explain RASP's computational model as well
* Section 5 is an interesting study on the effects of compression, though (as noted in Weaknesses below) I am not sure about the relevance or significance of these results.


**Weaknesses:**

* The paper does not sufficiently discuss the accuracy of the compiled programs (Section 3). Line 49 says "Any elementwise operations in RASP can be approximately computed by an MLP layer", and line 119 says that the MLPs and attention blocks "approximate arbitrary functions". Having error in the compiled programs is a perfectly reasonable limitation, but it would be very good to know the fidelity of these approximations.
* The paper assumes a high level of familiarity with prior work, which could be inlined into the paper.
  * Lines 59-62: I'm not familiar with Elhage et al. (2021)'s concept of a residual stream, but this seems to be a core concept in the paper. It would help to clarify these concepts and definitions in the paper. Appendix B.2 provides some information here, but this is still a bit insufficient (a "residual stream" is never actually defined).
  * In Section 3, it could also help to include a figure with RASP's syntax to understand the specific functions that must be compiled (seemingly, `select`, `selector_width`, `aggregate`, and the element-wise operations; `selector_width` isn't mentioned until Section 4.2), and the specific operations that are not yet supported by Tracr (e.g., those on Line 187). Again, Appendix B.3 helps, but is still not quite sufficient (e.g., it also does not define `selector_width`).
* Section 5 reads as an entirely different paper, with its own significant limitations. As the authors note, "even with a fairly restrictive compression setup, compressed models may not stay faithful to the original RASP program". This is certainly an interesting finding (and has ramifications on compression techniques broadly). But, given that these compressed Transformers neither follow the original RASP program, nor is it clear that this the behavior observed when compressing non-Tracr Transformers, it's not clear what to take away from these results.


**Questions:**

* How precisely do the compiled Transformers implement the original algorithms?
* Regarding "Disallow[ing] arbitrary selector combinations" in Appendices C and G: does this restriction reduce the set of programs that it is possible to represent with RASP?
* This is more of a curiosity than a criticism, but in Section 5.1 when projecting out of the compressed space, is there a reason to apply W^T rather than say the psuedoinverse of W?


**Limitations:**

The authors adequately discuss limitations in the appendix.

---

> ### Author Rebuttal · Authors · 2023-08-07
>
> > How precisely do the compiled Transformers implement the original algorithms?
>
> Tracr can compile any algorithm to a finite model that implements it exactly, i.e., with zero approximation error. This is because we know the full (discrete) input vocabulary for the model at compile-time. So, while some functions are implemented “approximately” by MLPs, we ensure that for all values that can occur for inputs from the vocabulary, the MLP approximation is exact (this is also briefly discussed in Appendix E.1). We will add a sentence clarifying this in the main body.
>
>
> > Regarding "Disallow[ing] arbitrary selector combinations" in Appendices C and G: does this restriction reduce the set of programs that it is possible to represent with RASP?
>
> Strictly, yes, this reduces the set of programs that we can compile. For example, without selector combinations, we cannot compile the sort program by Weiss et al. 2021 because it uses a combined selector to handle duplicates in the input:
>
> ```
> select(keys, keys, <) or (select(keys, keys, ==) and select(indices, indices, <)
> ```
>
> However, any such program can be refactored into an equivalent RASP program that we can compile. We describe this procedure at the end of Appendix G, which also discusses the technical reasons for disallowing arbitrary selector combinations. In brief: this is primarily because combined selectors a) produce large and inefficient models, and b) would break the correspondence between selectors and attention heads. We therefore leave doing this refactoring to users to avoid surprises in compilation.
>
> In practice, we have not found that this restriction introduces practical difficulties.
>
> Thinking about this question again helped us clarify our thinking on this issue; thank you – we will update the paper and the appendix to reflect that.
>
>
>
> > This is more of a curiosity than a criticism, but in Section 5.1 when projecting out of the compressed space, is there a reason to apply W^T rather than say the psuedoinverse of W?
>
> We made this choice primarily to be consistent with Elhage et al. 2021. Using the pseudoinverse of W would also be a reasonable choice.
>
>
> > The paper assumes a high level of familiarity with prior work, which could be inlined into the paper.
>
> Thanks for flagging this! We aim to address the insufficient discussion of necessary background by moving Appendix B.2 to a dedicated background section and improving it. We will also improve our discussion of RASP syntax to make the paper more self-contained.

---

> > ### Comment · Reviewer_XH5L · 2023-08-18
> >
> > Thanks to the authors for the detailed response. The authors have given strong answers to my remaining questions about the paper, so I'm raising my score by a point.

---

### Official Review · Reviewer_PkEE · 2023-07-05

**Soundness:** 4 excellent
**Presentation:** 4 excellent
**Contribution:** 4 excellent
**Rating:** 7
**Confidence:** 4

**Summary:**

This paper presents Tracr, a compiler that can take 'program' specifications and translate them into GPT (decoder only) style transformer models. Tracr is built on the RASP 'programming' language introduced by Weiss et. al., and translates a RASP program into model weights via an intermediate representation termed craft. The entire compilation step occurs first by translating a RASP program into a computational graph where operations in the program correspond to nodes in the graph. The RASP language is equipped with: sequence operations and selectors, and instructions: element-wise and select-and-aggregate operations, which correspond to the MLP layers and attention operation respectively. The nodes in the graph are then translated to MLPs and attention heads which are then assembled into a complete model. The authors show how to use Tracr in two examples: sorting and token counting. Lastly, they use RASP to examine the influence of various model compression schemes on the model's logic.

**Strengths:**

**Originality**\
The key previous work is the RASP paper by Weiss et. al., however, this work significantly expands the scope of that previous paper into a usable compiler for assessing the effectiveness of interpretability methods on decoder-only transformer models. I find the series of choices made in the program translation process to also be intuitive. This paper presents an intriguing new tool will be useful for evaluating interpretability methods.

**Quality/Clarity**\
The paper is quite well written and the figures help demonstrate the point of the work. The use of the is_x example was very helpful in following the compilation steps that tracr implements. This work is of high quality and delivers a useful tool for the purpose for which it sets out.

**Significance**\
It is hard to judge significance of a paper, but I'd hazard a guess that this paper opens up a new line of work  for testing mechanistic interpretability methods, which are a new increasingly popular method for reverse-engineering large-scale models. Personally, I am quite skeptical of most research in mechanistic interpretability as I think it is susceptible to just reading tea leaves, so the tracr compiler could end up taking the place of unit tests for new mechanistic interpretability schemes. Imagine compiling a program to a set of weights and then asking whether some new mechanistic interpretability technique can identify the underlying mechanism. Overall, this paper has interesting ideas that can likely be adapted in future work.





**Weaknesses:**

Overall, I think this work is an important one, and opens several questions that could be follow-up work. None of the weaknesses I discuss below are disqualifying.

I'll start with my major qualms with this work:

**How different are tracr model weights from those that gradient descent learns?**: The model weights that one gets from tracr are going to be quite different from those that you get when you collect data from the RASP function and simply learn a model on that data. And I just don't mean that the exact values are different, but the weights could encode qualitatively different behaviors. It seems to me that based on the rules here, tracr models will have sparser weight matrices than traditional models learned via SGD. Since interpreting a sparse model is easier than a dense one, do the authors see problems with tracr models being quite different from learned ones?

**Details**: As expected, a paper like this is packed with insights, and the details here actually matter. Specifically, I am referring to the heuristics used to 1) decide values that an s-op will take, 2) combining layers. Of course one has to make concrete choices here, but it is a bit unclear to me how these rules should impact the 'quality'/type of function you get out of tracr. Similarly, the last step, a crucial one, is also under explained in the main draft. Specifically, it is unclear what the authors mean by "factor", and exactly how they arrive at the specific values for these weight matrices. Some of this information is discussed in the appendix.

**The interpretability assessment pitch**: A main motivation of this work is to test interpretability methods. However, this paper stops short of that goal. It would've been amazing for the authors to apply a mechanistic interpretability to say a dyck example to see whether these methods can recover such logic. Even further, while the tool is an important component of testing an interpretability method, it is unclear to me how you would actually use it to do so. Section 5 attempts to do this, but not in a straightforward manner.

**Minor Issues**
1) I had to go read Elhage et. al. to understand what a 'residual stream' is. Is this interpretation now standard in the literature? Section B.2 in the appendix is actually important background notation.
2) Defn of mechanistic interpretability in lines 54-55 is circular. What is a mechanistic explanation? I think the second sentence is actually the real definition?

**Questions:**

See weaknesses section for a longer discussion of the key questions.



**Limitations:**

There is extensive discussion of various limitations in the Appendix and Conclusion section.

---

> ### Author Rebuttal · Authors · 2023-08-07
>
>
> > How different are tracr model weights from those that gradient descent learns?
>
> Your description is accurate: Tracr models are significantly sparser than real trained transformers, and, in particular, small Tracr models tend to be easy to interpret. There are straightforward ways to "obfuscate" the compiled models, e.g., by rotating their representations. Doing this makes the models significantly harder to understand. However, as you mentioned, Tracr models' weights look qualitatively different than the weights of compiled models. This is what we mean by Tracr models being “unrealistic” in Appendix A.2 when discussing limitations; we will make sure to clarify this in the main paper.
>
> This observation motivates the second part of our paper on compressing Tracr models using SGD. We find that this compression procedure makes the models significantly more realistic while maintaining the "ground truth" computation. There is some trade-off between models being natural and models having a ground truth interpretation (e.g. see Section 5.2). But we primarily see Tracr models filling the role of "unit tests" for mechanistic interpretability (Thanks for this metaphor!). So, we think it is acceptable to sacrifice some amount of "naturalness" for having the ground truth interpretation.
>
>
> > Specifically, it is unclear what the authors mean by "factor", and exactly how they arrive at the specific values for these weight matrices.
>
> We assume you are referring to the sentence “we then factor the W_QK and W_OV matrices into separate W_q, W_k, W_o, W_v weight matrices” in lines 142 and 143.
>
> In our implementation, we simply set $W_q = W_{QK}$ and $W_k = I$ as an identity matrix and $W_o = W_{OV}$ and $W_v = I$ as an identity matrix. However, we might use more sophisticated factorisations in the future. We agree this deserves some more details in the main paper.
>
>
> > I had to go read Elhage et. al. to understand what a 'residual stream' is. Is this interpretation now standard in the literature? Section B.2 in the appendix is actually important background notation.
>
> We agree that Section B.2. contains important background, and we aim to move it to the main paper when updating the paper.
>
>
> > The interpretability assessment pitch: A main motivation of this work is to test interpretability methods. However, this paper stops short of that goal. It would've been amazing for the authors to apply a mechanistic interpretability to say a dyck example to see whether these methods can recover such logic. Even further, while the tool is an important component of testing an interpretability method, it is unclear to me how you would actually use it to do so. Section 5 attempts to do this, but not in a straightforward manner.
>
> We agree that testing an interpretability method with a Tracr model would make the impact much more apparent. We are addressing this in follow-up work, but we felt it would have expanded the scope of this paper to the point that the contributions would have been harder to communicate clearly.

---

> > ### Comment · Reviewer_PkEE · 2023-08-10
> > **Concerns addressed**
> >
> > Hello,
> >
> > I have read the rebuttal, and I am satisfied with the response. I think the tracr library is a nice contribution, and should lead to important insights on the effectiveness of mechanistic interpretability methods.

---

### Official Review · Reviewer_767m · 2023-07-05

**Soundness:** 3 good
**Presentation:** 4 excellent
**Contribution:** 3 good
**Rating:** 7
**Confidence:** 3

**Summary:**

In this submission, the authors present Tracr, a compiler for RASP (a DSL for transformer computations) into transformer weights. The authors introduce their compilation approach, which includes an “assembly” language called Craft, which is used to represent the transformer weights agnostic to explicit implementations. They provide example models produced by Tracr (counting tokens & sorting, and refer to more examples in the repo). Most interestingly, they provide a way to compress these compiled models using SGD to study more involved topics in transformer explainability research, such as superposition.

**Strengths:**

- The compilation (section 3) is exceptionally well explained
- Their approach is especially powerful as a didactic tool
- The approach can serve as a guide for explainability research, and in providing a “ground truth” for explainability research
- The open source implementation can serve as a starting point for many researchers interested in explainability of transformer models
- Section 5 on compression is interesting and circumvents some of the limitations (otherwise it would be impossible to do research based on the compiled models on more involved concepts)

**Weaknesses:**

- Examples feel toyish; the limited size of programs (see their limitation section)
- That the compiled models could serve as a “ground truth” has to be taken with a grain of salt: Trained transformers, of course, optimize towards a different objective and solve a different problem.


**Questions:**

- How do the authors try to approach the mentioned limitations (in their last section) in future work? What is the path from here?
- Can the authors elaborate more on the choice of compression (page 6, line 224+)? Can you provide more detail how this would affect the assumption that it can serve as a “ground truth”?

**Limitations:**

See their limitations section and Appendix A2. I feel that the authors have honestly addressed the limitations.

---

> ### Author Rebuttal · Authors · 2023-08-07
>
> > How do the authors try to approach the mentioned limitations (in their last section) in future work? What is the path from here?
>
> There are technical limitations of Tracr that mostly come from design choices we made for simplicity (e.g., Tracr models embed different variables orthogonally, which can be quite inefficient). These limitations can be overcome with more sophisticated features. We plan to develop such features as they are needed for specific applications and we will encourage the research and open-source communities to contribute to Tracr.
>
> The RASP language has some limitations, such as using only binary attention patterns. It is not fully understood which of these limitations are most severe, and we are excited about recent and ongoing work that studies the expressivity of different models of transformer computations (e.g., for binary attention patterns, Merrill et al. 2022 is relevant). In future versions of Tracr, we might extend RASP to remove some of these limitations.
>
> Finally, there are fundamental limitations to compiling models. Of course, we will never compile a fully-fledged language model with Tracr. Instead, we think compiled models will be useful as an intermediate step between analysing toy models and real learned models.
>
> For a more detailed discussion of Tracr’s limitations and possible future work, see Appendix A.2.
>
> > Can the authors elaborate more on the choice of compression (page 6, line 224+)? Can you provide more detail how this would affect the assumption that it can serve as a “ground truth”?
>
> We aimed to find a compression procedure that:
> makes the compiled models more efficient and realistic;
> maintains the “ground truth”-property of the compiled models, i.e., that we can identify which part of the model implements which computation;
> is conceptually simple and natural.
>
> Requirements (1) and (3) suggest a gradient-descent-based compression is a natural choice. To achieve (2), we freeze most of the compiled weights and only learn a new embedding matrix. Limiting the compression to a single shared matrix W ensures that the model still has the same structure. In most cases, we still fully understand the compressed models after investigating the learned W (e.g. see Section 5.2), which suggests that the compressed models are still useful as a ground truth. However, sometimes the learned compression can be quite complex, making the model more difficult to use as a ground truth (e.g. see Section 5.3).
>
> We could try many other possible compression procedures here (and we might explore some in the future). But we think the proposed approach is a good trade-off between the three requirements and a natural starting point for studying compressing the compiled models. There is a fundamental trade-off between having a ground truth understanding of the model and having a very natural-looking model. While we provide a preliminary investigation of this in the paper, there is undoubtedly much room for future work here.

---

> > ### Comment · Reviewer_767m · 2023-08-18
> >
> > I thank the authors for their response and for explaining the decisions; that made it clearer. I don't have any more questions.

---

### Author Rebuttal · Authors · 2023-08-07

We thank all reviewers for their insightful comments. We are glad the reviewers found our paper clear and appreciated the contribution of Tracr to studying the computational models of transformers (Reviewers XH5L, ACvV), to advancing interpretability research (Reviewers 767m, PkEE), and as a didactic tool (Reviewer 767m).

We respond to any open questions in the individual responses to the reviewers, and we hope this addresses any remaining uncertainties about our paper. We will be available during the discussion period if any further questions arise.

---

### Decision · Program_Chairs · 2023-09-21

**Decision:**

Accept (spotlight)

**Comment:**

The paper gives a method for compiling human-readable programs into decoder-only transformers. The structure of the compiled models is used to design experiments (for example, a study of superposition in transformers) and to evaluate interpretability methods.

The reviewers were uniformly positive about both the paper's contributions and the presentation. There are some questions about how broadly the methods here are applicable to transformers that implement natural tasks; however, the paper is above the bar in spite of these questions. Given this, I am enthusiastically recommending acceptance. Please make sure to incorporate the reviewers' feedback carefully in the final version.